# Calorie Restriction Suppresses Premature Ageing in Pro-Apoptotic Yeast Mutants Through an Autophagy-Independent Mechanism

**DOI:** 10.3390/ijms27010464

**Published:** 2026-01-01

**Authors:** Benedetta Caraba, Mariarita Stirpe, Vanessa Palermo, Alessia Ayala Alban, Arianna Montanari, Michele Maria Bianchi, Claudio Falcone, Cristina Mazzoni

**Affiliations:** 1Department of Biology and Biotechnologies “C. Darwin”, Sapienza University of Rome, Piazzale Aldo Moro 5, 00185 Roma, Italymariaritastirpe@gmail.com (M.S.); vanessa.palermo76@gmail.com (V.P.); ayala.alessia@gmail.com (A.A.A.); ari.montanari@uniroma1.it (A.M.); michele.bianchi@uniroma1.it (M.M.B.); claudio.falcone48@gmail.com (C.F.); 2Water Research Institute, National Research Council (IRSA-CNR), 00010 Rome, Italy

**Keywords:** autophagy, aging, calorie restriction, *LSM4*

## Abstract

The budding yeast *Saccharomyces cerevisiae* has long served as a valuable model for investigating the molecular mechanisms underlying aging. Calorie restriction (CR) is a well-established intervention that extends lifespan across species, yet the underlying molecular mechanisms remain incompletely understood. In this study, we examined the effects of CR on the chronological lifespan, oxidative stress response, and autophagic activity of the *Saccharomyces cerevisiae* mutant *Sclsm4Δ1*, which exhibits premature aging and elevated reactive oxygen species (ROS) levels due to defects in mRNA decapping and processing-bodies (PB) dynamics. We found that both moderate (0.1% glucose) and extreme (water incubation) CR significantly extended the lifespan of *Sclsm4Δ1* mutants and markedly reduced intracellular ROS accumulation without activating autophagy. These findings indicate that the beneficial effects of CR stem from improved redox homeostasis and metabolic adaptation, rather than from canonical autophagic pathways. Similar protective effects were observed in a chromosomal *lsm4Δ1* mutant generated via CRISPR–Cas9, confirming that CR rescues aging-related phenotypes in different genetic backgrounds. These insights reinforce the roles of nutrient signaling, RNA metabolism, and redox balance in lifespan regulation, offering new perspectives on the conserved anti-aging effects of calorie restriction.

## 1. Introduction

In recent decades, the yeast model system *Saccharomyces cerevisiae* has been used to study the complex and dynamic process of aging, which poses challenges because of the intricate pathways and networks involving both genetic and environmental factors. However, selecting the most suitable yeast strain for analysis is crucial, as phenotypes of interest may vary across different genetic backgrounds [1,2,3]. Our research group has characterized a mutant strain involved in the decapping process, which exhibits premature onset of aging-related phenotypes, such as nuclear fragmentation and accumulation of reactive oxygen species (ROS), along with a notably short lifespan during chronological aging, defined as the survival rate of stationary cultures [4,5,6]. This particular yeast strain is a *S. cerevisiae* strain that expresses a truncated form of the *Kluyveromyces lactis* (*Kllsm4Δ1*) or *S. cerevisiae* (*Sclsm4Δ1*) *LSM4* essential gene on a centromeric plasmid, whereas the endogenous form is under the control of the *GAL1-GAL10* promoter and is therefore repressed by glucose and induced by galactose. Lsm4 is a component of Lsm complexes located in the nucleus or cytoplasm, where it regulates mRNA splicing, stability, and translation [7]. Although splicing regulation remains intact owing to the preservation of the two N-terminal Sm-like domains, truncation compromises mRNA turnover, as removal of the C-terminal Q/N-rich domain prevents efficient recruitment to PBs [8].

Owing to their heightened sensitivity to stress, premature senescence, and increased cell death, *lsm4Δ1* mutants represent an effective experimental system for elucidating the molecular networks that regulate longevity and stress resistance. For these reasons, *lsm4Δ1* mutants have been widely employed as tester strains to investigate the cellular consequences of expressing human genes [9,10] and to evaluate the anti-aging potential of natural bioactive compounds [11,12,13], highlighting the versatility of yeast as a model for recapitulating human cellular processes and screening molecules with potential therapeutic relevance. The phenotypic defects of *Kllsm4Δ1* mutants can be suppressed by the overexpression of yeast genes such as *HIR1*, *PGK1*, and *NEM1*, suggesting that these defects are linked to alterations in chromatin structure and accessibility, energy metabolism, lipid homeostasis, and autophagy [14,15,16].

In a recent study, we underscored the sensitivity of a strain expressing the truncated form of *S. cerevisiae LSM4* (*Sclsm4Δ1*) to various stimuli, such as caffeine and rapamycin treatment and nitrogen starvation, indicating that the Lsm complex plays a vital role in regulating autophagy [6]. It is now recognized that *ATG* mRNAs, which are involved in every step of autophagy in yeast, can be regulated post-transcriptionally through the combined action of the nutrient-sensing factor TORC1 and the decapping complex Dcp2-Dhh1 [17]. In nutrient-rich conditions, such as in the presence of glucose, amino acids, and other nitrogen sources, autophagy is suppressed by promoting the degradation of *ATG* mRNAs. This degradation occurs through phosphorylation by TORC1, which subsequently activates Dcp2, leading to the decapping and degradation of mRNAs via Xrn1 exoribonuclease activity. Conversely, the high phosphorylation activity of TORC1 promotes anabolic responses, including ribosome biogenesis and activation, protein synthesis, as well as lipid and nucleotide synthesis, which in turn foster cellular growth and division [18,19].

Gatica and colleagues [20] detailed the unconventional role the Pat1/Lsm complex in activating autophagy by showing that under nitrogen deprivation, it associates with certain *ATG* gene mRNAs to protect their trimmed poly-A tail from further degradation, ensuring their accumulation and robust autophagic response, whereas mutations in *LSM1* or *PAT1* genes significantly impair this activation under the same starvation conditions [20].

Both nitrogen and glucose starvation inhibit TORC1 activity, but they affect different stages and effectors within the signaling cascade. Elevated amino acid levels activate TORC1 via the action of small GTPases Gtr1 and Gtr2, orthologs of mammalian Rag GTPases, where the active conformation promotes TORC1 localization to the vacuolar membrane [21]. Conversely, in the absence of nitrogen sources, the SEA Complex (SEAC), homologous to mammalian GATOR1/2, targets Gtr1/2 regulatory subunits to induce GTP hydrolysis, preventing TORC1 activation [22]. Glucose starvation triggers AMP-activated protein kinase (AMPK Snf1), which decreases TORC1 activity by causing the dissociation of its components, particularly the disassembly of Kog1 (RAPTOR in mammals), the primary subunit responsible for targeting and recruiting substrates for Tor kinases (Tor 1/2) [23].

In recent years, the intriguing link between TORC1 activity and aging has drawn significant attention from the scientific community, leading to numerous studies aimed at uncovering the role of the kinase complex in lifespan extension. Research conducted on animal and cellular models has demonstrated that TORC1 inhibition, achieved either through specific drugs like rapamycin or by inducing nutrient and energy stress, contributes to lifespan prolongation [24,25], with the activation of the autophagy process proposed as a key factor in this effect [26].

One intriguing method to extend the lifespan of various model systems, ranging from yeast to metazoans, involves initiating the autophagy process through calorie restriction (CR) [27], facilitated by the activation of AMPK Snf1 and Sir2 (Sirt1 in humans). In yeast, the Silent Information Regulator 2 (*SIR2*) acts as a histone deacetylase, playing a crucial role in cellular homeostasis by inhibiting rDNA recombination, silencing basal transcription at telomeres, and regulating the expression of the mating-type locus, with its activity closely linked to the cell’s oxidative state. During calorie restriction, the concentration of its cofactor NAD+ becomes sufficient to enhance its deacetylation activity, thereby repressing the toxic accumulation of extrachromosomal rDNA circles (ERCs) and promoting cellular health, ultimately leading to an increased lifespan [28]. Furthermore, several studies have demonstrated its role in activating autophagy under these conditions, positively impacting the organism’s lifespan [29,30,31].

To gain a deeper understanding of the role of mRNA regulation and degradation in the autophagic process, we extended our research on yeast decapping mutants under various starvation conditions, specifically examining the energy stress caused by limited glucose availability. Our findings revealed that CR alleviated apoptotic phenotypes and premature aging in *Sclsm4Δ1* mutant strains. However, lifespan extension appears to be linked to redox homeostasis and is not dependent on autophagy activation, although the underlying molecular mechanisms remain elusive.

## 2. Results

### 2.1. Calorie Restriction Elongates Chronological Lifespan and Reduces ROS Accumulation of Sclsm4Δ1 Mutant Strain Without Altering Its Gene Expression

An effective method for extending yeast lifespan is to reduce the glucose concentration in the culture medium [27,30]. Consequently, we examined the lifespan of both wild-type (CML39-11A) and *Sclsm4Δ1* mutant strains under various media conditions to confirm this behavior. Yeast and bacterial strains used in this work are reported in Table 1 and Table 2, respectively. Compared to the wild type strain (Figure 1A), the *Sclsm4Δ1* mutant strain, showed in Figure 1C, exhibited a significantly shorter lifespan in nutrient-rich conditions (SD 2% glucose, in blue), corroborating previous findings [6]. Upon transitioning to CR (SD 0.1% glucose in orange) and extreme calorie restriction (S-glu in green and pure water in yellow), the viability of both strains remained high, with minimal variations in S-glu and water conditions, whereas in SD 0.1% it remained relatively stable throughout the experiment. Minor variations were also observed in the cellular absorbance of the wild type (Figure 1B), indicating the absence of cell lysis. In contrast, the *Sclsm4Δ1* mutant showed some loss of OD in S-glu media (Figure 1D), suggesting approximately 20% cell lysis after three days of growth under these conditions.

One potential reason for the extended lifespan under these conditions could be a change in the cell’s oxidative state. The *Sclsm4Δ1* mutant strain, in particular, demonstrated an accumulation of ROS, leading to mRNA oxidation, heightened sensitivity to oxidative stress, and consequently, a reduced lifespan [4,5,6,32]. To investigate further, we examined the number of ROS-positive cells under various conditions using fluorescence microscopy of DHR 123 stained cells. As illustrated in Figure 2B, the *Sclsm4Δ1* mutant strain in the exponential phase under nutrient-rich conditions (depicted in blue) showed about 55% ROS-positive cells, in stark contrast to the wild-type strain, which had fewer than 10% ROS-positive cells. Under calorie-restricted conditions, the *Sclsm4Δ1* mutant exhibited a significant decrease in ROS-positive cells, with *p*-values < 0.0001 across all tested conditions, compared to the SD 2% glucose control. Conversely, the wild-type strain did not show a statistically significant reduction in ROS-positive cells under SD 0.1% glucose, and under complete glucose starvation (S-glu, 0% glucose), there was even a slight increase in ROS-positive cells.

The MCY4/*Sclsm4Δ1* strain contains the full-length *LSM4* gene regulated by the *GAL1–GAL10* promoter, which is suppressed in the presence of glucose. To confirm that the observed lifespan extension and reduction in intracellular ROS in the *Sclsm4Δ1* mutant under calorie restriction were not due to reactivation of the full-length *LSM4* gene, we measured its mRNA levels using real-time PCR. We employed primers targeting the C-terminal region of *LSM4* (ScLSM4 C-term Fw and ScLSM4 C-term Rev; Table 3), which is absent in truncated *Sclsm4Δ1*. As illustrated in Figure 3A, the mRNA levels under calorie-restricted (SD 0.1%) and extreme calorie-restricted (S-glu, H_2_O) conditions were similar to those in glucose-repressed medium (SD), showing no statistically significant differences, except for S-glu. In fact, under these conditions, a slight activation of the full-length *LSM4* gene could be observed. Anyway, all conditions exhibited a highly significant reduction in full-length *LSM4* mRNA compared to that in galactose-grown cells, where the *GAL* promoter was active. These findings confirm that the extended lifespan and enhanced cellular fitness under calorie-restricted conditions are not due to the reactivation of the full-length *LSM4* gene.

To evaluate the expression of the truncated *Sclsm4Δ1* mRNA, we used primers targeting the N-terminal region shared by both the full-length and truncated genes (ScLSM4 N-term Fw and ScLSM4 N-term Rev; Table 3). As shown in Figure 3B, although variations were noted among the samples, no statistically significant changes were detected across the tested conditions. This suggests that calorie restriction does not lead to either overexpression or downregulation of the *Sclsm4Δ1* truncated form.

### 2.2. Calorie Restriction Does Not Induce the Accumulation of Autophagosomal Structures

In our previous study, we demonstrated that the *Sclsm4Δ1* mutant strain showed impaired processing of the GFP-Atg8 fusion protein, a well-established autophagy marker, under nitrogen starvation, leading to a significant accumulation of cytoplasmic GFP-positive puncta [6]. As illustrated in Figure 4, glucose starvation did not effectively facilitate the cleavage of GFP from the fusion reporter in either the wild-type strain (CML39-11A; panels A and B) or the *Sclsm4Δ1* mutant (panels C and D). Only a minimal amount of free GFP was detectable in the wild-type strain under glucose-rich conditions, with no substantial increase observed under glucose restriction.

Fluorescence microscopy analysis (Figure 5) indicated a reduction in GFP-positive cytoplasmic foci in the *Sclsm4Δ1* mutant after 16 h of growth under moderate glucose restriction (SD 0.1%), as depicted in Figure 5D. In contrast, the number of puncta remained consistent under conditions of glucose abundance and complete glucose deprivation (S-glu). A similar trend was observed in the wild-type strain, and after three days of culture, the number of cytoplasmic GFP dots further decreased in both strains under SD 0.1% conditions (Figure 5F). These findings suggest that glucose restriction may lead to a physiological reduction in autophagosomal structures rather than a robust activation of autophagic flux, at least in this genetic background.

### 2.3. Calorie Restriction Elongates the Chronological Lifespan of the Genomic lsm4Δ1 Mutant Strain

In the MCY4/*Sclsm4Δ1* strain, the *LSM4* gene was integrated into the chromosome under the control of the *GAL1-GAL10* promoter. Although we demonstrated that the chromosomal copy of the gene is not activated during CR, using CRISPR–Cas9 genome editing, we constructed a chromosomal copy of *Sclsm4Δ1* fused to GFP. This truncated chromosomal version of *LSM4* was under the control of its native promoter. The corrected integration and expression of the fusion protein *Sclsm4Δ1*-GFP were verified by PCR and Western blotting, respectively (Appendix A). This strain exhibited phenotypes comparable to those observed in MCY4/*Sclsm4Δ1* [4,5,6], including nuclear fragmentation, accumulation of reactive oxygen species (ROS) during aging, and sensitivity to acetic acid, caffeine, and rapamycin (Appendix A). An isogenic strain expressing the full-length *LSM4*–GFP fusion was used as a control. Although these phenotypic alterations were less pronounced in this background compared to MCY4/*Sclsm4Δ1*, cells still displayed a significantly shorter chronological lifespan than the wild type (Figure 6A). Notably, both calorie restriction (SD medium containing 0.1% glucose) and incubation in water restored the reduced lifespan, indicating that these conditions effectively counteracted the *Sclsm4Δ1* aging phenotype across different yeast genetic backgrounds (Figure 6B).

## 3. Discussion

In this study, we investigated the effects of calorie restriction (CR) on the chronological lifespan, oxidative stress response, and autophagic activity of the *Sclsm4Δ1* mutant strain of *Saccharomyces cerevisiae*. Our findings demonstrate that CR significantly extends the chronological lifespan of the *Sclsm4Δ1* mutant and reduces intracellular reactive oxygen species (ROS) accumulation without altering autophagy or *LSM4* gene expression. Our data suggest that the beneficial effects of CR arise from global physiological adaptations that mitigate oxidative damage rather than autophagy activation or transcriptional modulation of the genomic *LSM4* expression.

### 3.1. Calorie Restriction Mitigates Oxidative Stress and Extends Lifespan in the Sclsm4Δ1 Mutant

The *Sclsm4Δ1* mutant exhibits premature aging, increased ROS accumulation, and stress sensitivity, likely associated with defects in mRNA decapping and P-body dynamics. Under both moderate (0.1% glucose) and extreme (S-glu or water incubation) calorie restriction, this strain showed a robust extension of chronological lifespan, accompanied by a pronounced decrease in ROS-positive cells compared with glucose-replete conditions. Under extreme calorie restriction conditions, a modest level of cell lysis was observed, reaching approximately 30% after 10 days, exclusively in the S-glu medium. Although premature lysis can contribute to a reduction in lifespan, a certain degree of programmed cell death remains essential for population fitness: it eliminates damaged or non-viable cells and releases metabolites that can be reutilized by healthier cells, ultimately allowing the fittest fraction of the population to survive longer [33].

Notably, ROS reduction was significant in the mutant (*p* < 0.0001) but not in the wild-type strain, indicating that CR preferentially benefits cells with compromised RNA-processing machinery. These results are consistent with previous studies showing that lifespan extension in yeast under CR is linked to improved redox homeostasis and reduced mitochondrial superoxide production through the modulation of metabolic fluxes and activation of stress response pathways, such as Snf1/AMPK, *SIR2*, and *TOR* signaling [34].

Quantitative PCR analysis confirmed that *LSM4* mRNA levels remained unchanged under both moderate and extreme CR, excluding the possibility that lifespan extension results from the reactivation of the full-length gene under the control of the *GAL1–GAL10* promoter. This indicates that CR acts independently of *LSM4* transcription, likely by modulating the metabolic and oxidative balance. Thus, nutrient limitation appears to compensate for the deleterious consequences of *Sclsm4Δ1* truncation through mechanisms that reduce oxidative stress and enhance cellular fitness.

### 3.2. Calorie Restriction Does Not Activate Autophagy in Either Wild-Type or Mutant Strains

We previously reported that the *Sclsm4Δ1* mutant is unable to complete autophagy under nitrogen starvation, and this condition, as well as autophagy inducers such as rapamycin, is very toxic to this strain [6]. Our data indicate that CR does not lead to the accumulation of autophagosomal structures in wild-type or *Sclsm4Δ1* strains. Fluorescence microscopy revealed a decrease in cytoplasmic GFP-positive puncta after 16 h of glucose restriction (SD 0.1%), and this reduction persisted after prolonged incubation (three days). One possibility is an increase in autophagic flux, in which newly formed autophagosomes are rapidly delivered to and degraded within the vacuole, preventing their accumulation in the cytoplasm. Nevertheless, Western blot analysis using GFP–Atg8 as a reporter revealed no significant cleavage of GFP under conditions of glucose restriction or starvation, suggesting that autophagic flux was not strongly induced. These observations suggest that 0.1% glucose CR may promote physiological stabilization of the cytoplasm rather than a robust activation of autophagy, indicating that the protective effects of CR on lifespan and ROS accumulation are independent of canonical autophagic processes.

### 3.3. Calorie Restriction Restores Lifespan in the Genomic Sclsm4Δ1 Mutant

To confirm that the observed effects were not limited to the plasmid-based system, we generated a chromosomal version of the *Sclsm4Δ1* mutation fused to GFP using CRISPR–Cas9 in a different genetic background. This strain reproduced the phenotypes of the MCY4/*Sclsm4Δ1* background, including nuclear fragmentation, elevated ROS accumulation, and sensitivity to stress-inducing agents. Although the phenotypes were milder in this genetic background, the mutant still exhibited a significantly shorter chronological lifespan than the wild type. Remarkably, both calorie restriction (SD 0.1% glucose) and incubation in water completely rescued this defect, restoring the lifespan to near wild-type levels. These findings confirm that CR exerts a broad protective effect on *lsm4Δ1-*related aging phenotypes across different genetic contexts.

### 3.4. Concluding Remarks

Taken together, our results reveal that calorie restriction promotes longevity and oxidative stress resistance in *Sclsm4Δ1* mutant strains through mechanisms independent of *LSM4* reactivation or autophagy induction. The data suggest that metabolic adaptation to low nutrient availability enhances redox homeostasis and cellular maintenance processes, thereby compensating for the molecular deficiencies associated with the loss of the Lsm4 C-terminal domain. These findings underscore the interplay between RNA metabolism, energy sensing, and oxidative stress in the control of yeast chronological lifespan and provide further insight into the molecular pathways through which calorie restriction exerts conserved anti-aging effects.

Our study aligns with the well-established view that CR represents one of the most effective interventions to delay cellular aging and extend longevity across species, including yeast, nematodes, and mammals [27], and it fits into the intriguing scenario of research studies on the underlying cellular mechanisms [35,36,37,38]. The observation that lifespan extension in *Sclsm4Δ1* occurs independently of autophagy suggests that alternative stress response pathways, such as those controlling mitochondrial function, oxidative stress detoxification, or mRNA surveillance, may play a predominant role in mediating the beneficial effects of nutrient limitation. Increased mitochondrial functionality and oxidative stress response have been correlated with longevity in different studies, even in unconventional yeast species [39]. In fact, in a recent study [40], among the genes upregulated during calorie restriction in humans, the authors found *PDP2*, which encodes pyruvate dehydrogenase phosphatase catalytic subunit 2, a mitochondrial protein that enhances the utilization of pyruvate in oxidative phosphorylation involved in aging [41]. Moreover, in the same study, one of the upregulated genes upon CR was *PATL1*, a *PAT1* homolog 1 that coordinates the assembly and activation of a decapping messenger ribonucleoprotein (mRNP) and promotes 5′–3′ mRNA degradation [42].

Future studies will be aimed at dissecting the molecular circuitry connecting RNA processing defects to metabolic and redox signaling in the context of aging.

## 4. Materials and Methods

### 4.1. Yeast Strains, Growth Conditions and Plasmids Construction

The *Saccharomyces cerevisiae* strains used in this study are listed in Table 1. Cell cultivation occurred at 28 °C in YPD medium, which consists of 1% yeast extract (Gibco, Thermo Fisher Scientific, Inc., Waltham, MA, USA, #212750), 2% bacto-peptone (Gibco, Thermo Fisher Scientific, Inc., Waltham, MA, USA, #21167), and 2% glucose. Additionally, cells were grown in SD medium containing 0.67% yeast nitrogen base without amino acids (Becton, Dickinson and Company, 1 Becton Drive, Franklin Lakes, NJ, USA, #291940) and 2% glucose, supplemented with the necessary auxotrophic nutrients. For experiments inducing autophagy through calorie restriction, cells were cultured with auxotrophic supplements and methionine until they reached the post-diauxic phase (0.8–1.2 OD_600_). They were then washed with water and resuspended in water, S-glu (0.67% yeast nitrogen base without amino acids (Becton, Dickinson and Company, 1 Becton Drive, Franklin Lakes, NJ, USA, #291940)), and SD 0.1% (0.67% yeast nitrogen base without amino acids, 0.1% glucose), with added auxotrophic requirements, and methionine. After 16 h, the cells were collected and proteins were extracted. For survival tests during chronological life span (CLS) under calorie restriction conditions, cells were grown in SD until they reached the post-diauxic phase (0.8–1.2 OD_600_), washed with water, and then resuspended in water (at OD_600_ of 1,2 for the wild type and 1,5 for the MCY4/*Sclsm4Δ1* mutant strain), S-glu, SD, and SD 0.1% media (at OD_600_ of 0,8 for both the wild type and the MCY4/*Sclsm4Δ1* mutant strain), as previously described. Appendix A shows the OD_600_ on day1 for each strain and condition. S-gal medium (0.67% yeast nitrogen base without amino acids, 2% D-galactose) was used to induce the endogenous expression of *LSM4* in the MCY4/*Sclsm4Δ1* mutant strain. To obtain solid media, 2% Bactoagar was incorporated (Becton, Dickinson and Company, 1 Becton Drive, Franklin Lakes, NJ, USA, #214010).

*Escherichia coli* strain DH5α used for plasmid DNA propagation is listed in Table 2. Bacterial cells were grown at 37 °C in LB medium (0.5% yeast extract (Gibco, Thermo Fisher Scientific, Inc., Waltham, MA, USA, #212750), 1% tryptone (Gibco, Thermo Fisher Scientific, Inc., Waltham, MA, USA, #211705), and 0.5% NaCl) supplemented with 100 μg/mL ampicillin.

Plasmids pRS313/*Sclsm4Δ1* and pRS313/*ScLSM4* were obtained in our previous work [6,43].

Plasmid pUG36/ATG8 was generously provided by T. Eisenberg and his colleagues [44].

Plasmid pUG35-URA was a gift from J. H. Hegemann, Heinrich-Heine-Universitat, Dusseldorf, Germany.

Plasmids pCfB2312 and pCfB2311 were purchased through AddGene from Irina Borodina and colleagues [45]. Transformation of the selected strain was performed using the PEG/LiAc method according to Gietz and Woods [46].

Transformations involving pRS313/*Sclsm4Δ1*, pRS313/ScLSM4, and pUG36/ATG8 were performed using the ONE-STEP method. The transformation mixture consisted of ONE-STEP buffer, which included 40% PEG 3350, 0.2 M LiAc, 0.1 M DTT, and 0.1 μg/μL ssDNA carrier (Sigma-Aldrich, Darmstadt, Germany, D1626) [47].

**Table 1 ijms-27-00464-t001:** *S. cerevisiae* strains used in this study were as follows.

Strain	Genotype	Source
MCY4	MATα, *ade1-101*, *his3-Δ1*, *trp1-289*, *ura3*, *LEU-GAL1-SDB23*	[48]
MCY4/*Sclsm4Δ1*	MATα, *ade1-101*, *his3-Δ1*, *trp1-289*, *ura3*, *LEU-GAL1-SDB23* pRS313/*Sclsm4Δ1*	[6]
CML39-11A	MATa, *ade1-101*, *his3-Δ1*, *leu2*, *ura3*, *trp1-289*	[43]
MCY4/*ScLSM4*	MATα, *ade1-101*, *his3-Δ1*, *trp1-289*, *ura3*, *LEU-GAL1-SDB23* pRS313/*ScLSM4*	[6]
MCY4/*Sclsm4Δ1* pUG36/*ATG8*	MATα, *ade1-101*, *his3-Δ1*, *trp1-289*, *ura3*, *LEU-GAL1-SDB23* pRS313/*Sclsm4Δ1*, pUG36/*ATG8*	[6]
CML39-11A pUG36/*ATG8*	MATa, *ade1-101*, *his3-Δ1*, *leu2*, *ura3*, *trp1-289* pUG36/*ATG8*	[6]
BY4741	*Mat a*, *his3-Δ1*, *leu2-Δ0*, *met15-Δ0*, *ura3-Δ0*	[49]
BY4741 pUG36/*ATG8*	*Mat a*, *his3-Δ1*, *leu2-Δ0*, *met15-Δ0*, *ura3-Δ0* pUG36/*ATG8*	[44]
BY4741 LSM4-GFP	*Mat a*, *his3-Δ1*, *leu2-Δ0*, *met15-Δ0*, *ura3-Δ0*	Thermo-Fisher Yeast GFP Clone Collection [50]
BY4741 *lsm4Δ1-*GFP	*Mat a*, *his3-Δ1*, *leu2-Δ0*, *met15-Δ0*, *ura3-Δ0*, *lsm4Δ1::GFP*	This work

**Table 2 ijms-27-00464-t002:** *E. coli* strain used in this work.

Strain	Genotype	Source
DH5α	dlacZ Δ M15 Δ(lacZYA-argF) U169 recA1 endA1 hsdR17(rK-mK+) supE44 thi-1 gyrA96 relA1	[51]

### 4.2. Viability Assays

Beginning on the first day, 3 × 10^4^ cells were plated daily on a YPD-coated slide and examined under an optical microscope after incubating for 1–2 days at 28 °C. The percentage of cells forming microcolonies was used to determine cell viability [52].

### 4.3. Extraction of Total RNA, Synthesis of cDNA, and Real-Time qPCR for Analyzing mRNA Expression of ScLSM4

Cells of strain MCY4/*Sclsm4Δ1* were cultivated in 80 mL of SD medium and collected during the exponential growth phase (0.2–0.4 OD_600_). Subsequently, they were divided, rinsed with H_2_O, and resuspended in 20 mL of SD, SD 0.1%, S-glu, and S-gal, followed by incubation at 28 °C for 24 h. The following day, cells equivalent to 4 OD_600_ were washed with H_2_O, resuspended in 200 μL of lysis buffer (0.5 M NaCl, 0.2 M Tris-HCl pH 7.5, 10 mM EDTA, 1% SDS) and 200 μL of phenol-chloroform-isoamyl alcohol (PCI) 25:24:1 (Sigma-Aldrich, Darmstadt, Germany, 77617), and disrupted by vortexing with micro glass beads. After adding 300 μL of lysis buffer and 300 μL of PCI, the cells were centrifuged at 10,000 rpm for 5 min at 4 °C, and the supernatant was precipitated with three volumes of EtOH at −20 °C for 30 min. The precipitated nucleic acids were dissolved in 15 μL of RNase-free H_2_O. RNA integrity was assessed through electrophoresis on a 1% agarose gel in 1× TAE buffer (40 mM Tris, 20 mM Acetate, and 1 mM EDTA) stained with Ethidium Bromide. RNA was treated with DNaseI using the DNA-free TM kit (Invitrogen, Thermo Fisher Scientific, Inc., Waltham, MA, USA AM1906) and reverse transcribed into cDNA using the SensiFAST cDNA Synthesis Kit (Meridian Bioscience, Inc., Cincinnati, OH, USA, BIO-65053), according to the manufacturer’s instructions. To assess *LSM4* expression levels, the resulting cDNAs were used as templates for real-time qPCR, using the primers listed in Table 3 and the SensiFAST SYBR Hi-ROX kit (Meridian, Bioscience, Inc., Cincinnati, OH, USA, BIO-92020). The *TDH3* gene (glyceraldehyde-3-phosphate dehydrogenase) was used as a reference. Data were collected using Rotor Gene Q (Qiagen, Hilden, Germany) and analyzed using the ΔΔCt method. The results are shown in Figure 3.

**Table 3 ijms-27-00464-t003:** Primers used in real-time quantitative PCR for *ScLSM4* expression.

Primer Name	Oligonucleotide Sequence
*ScLSM4 C-term Fw*	5′-CCG CCG TCC ATA CTC TCA AA-3′
*ScLSM4 C-term Rv*	5′-TTG GAC GGA CCC ACC TAA AC-3′
*ScLSM4 N-term Fw*	5′-ATT GAC CAA CGT AGA TAA CTG GA-3′
*ScLSM4 N-term Rv*	5′-TAC GGC TTT ACT GCT CTC AG-3′
*TDH3 Fw*	5′-CGG TAG ATA CGC TGG TGA AGT TTC-3′
*TDH3 Rv*	5′-TGG AAG ATG GAG CAG TGA TAA CAA C-3′

### 4.4. Fluorescence Microscopy

Nuclear morphology was assessed using DAPI staining at a concentration of 1 μg/mL (Sigma-Aldrich, Darmstadt, Germany, D8417) on 1 mL of cells in the exponential growth phase (0.2–0.4 OD_600_) that had been fixed with 70% (v/v) ethanol. Reactive oxygen species (ROS) were identified by incubating 1 mL of cells with 5 μg/mL of DHR 123 (Sigma-Aldrich, D1054) for 4 h at 28 °C, followed by analysis under a fluorescent microscope (Axioskop 2, Carl Zeiss, Oberkochen, Germany). The formation and movement of autophagosomes were visualized using the reporter plasmid pUG36/ATG8 and examined using the same fluorescent microscope as described previously [3]. Starting from an exponential preculture in 20 mL of SD medium with auxotrophic requirements (0.2–0.3 OD600), cells were washed with water, resuspended in SD, SD 0.1%, and S-glu media, and then analyzed under a fluorescent microscope after 4 h (exponential phase), 16 h (post-diauxic phase), and 3 days (stationary phase). The percentage of cells positive for GFP-Atg8 dots was calculated from the total number of fluorescence-positive cells.

### 4.5. Assessment of Growth on Glycerol, and Sensitivity to Caffeine, Acetic Acid, and Rapamycin

For glycerol growth, caffeine, acetic acid, and rapamycin sensitivity tests, serial dilutions of the strains of interest were spotted on YPD, YPY, YPD + 0.25% caffeine, YPD + 60 mM acetic acid, and YPD + 6 nM rapamycin plates, and their viability was assessed after 2–3 days of incubation at 28 °C.

### 4.6. Protein Extraction and Western Blot Analysis

The strains of interest were cultivated in 20 mL of SD medium supplemented with auxotrophic requirements and methionine (10 μg/mL). They were harvested during the exponential phase (0.2–0.4 OD_600_), washed with H_2_O, and divided into 10 mL of SD medium with auxotrophic requirements and methionine (10 μg/mL), 10 mL of SD 0.1% with auxotrophic requirements and methionine (10 μg/mL), and S-glu medium with auxotrophic requirements and methionine (10 μg/mL). The flasks were incubated at 28 °C. SD-cultured cells were harvested at their logarithmic growth phase (Exp, 0.4 OD_600_) and post-diauxic phase (after 16 h, 0.9–1 OD_600_), while S-glu and SD 0.1% cultured cells were harvested after 16 h. Cells amounting to 2 OD600 were washed with H_2_O, resuspended in 200 μL of NaOH 2 M/β-mercaptoethanol 5%, and chilled on ice for 10 min. Protein precipitation was carried out with TCA at a final concentration of 8.3%, followed by centrifugation at 13,000 rpm for 15 min, and the pellet was suspended in 100 μL of loading buffer (50 mM Tris-HCl pH 6.8; 100 mM β-mercaptoethanol; 2% SDS, 0.1% bromophenol blue; 10% glycerol). Samples were then boiled at 95 °C for 5 min and loaded onto a 12% acrylamide SDS-PAGE gel. A protein marker was loaded into the first lane (Thermo-Fisher, Inc., Waltham, MA, USA, LC5925). The separated proteins were transferred onto a nitrocellulose membrane via electroblotting. Ponceau red staining was used as a loading control (0.1% Ponceau S (Sigma-Aldrich, Darmstadt, Germany, P-3504) and 5% acetic acid). Autophagic cargo processing was examined via immunoblotting analysis using an anti-GFP antibody (α-mouse-GFP, Santa Cruz Biotechnology, Dallas, TX, USA, sc-9996) to detect GFP-Atg8, as described previously [49]. The secondary antibody used was HRP-associated sc-2060 (Santa Cruz Biotechnology, Dallas, TX, USA) anti-mouse (goat). The percentage of autophagy activation was determined as the ratio between free GFP and total GFP (free GFP/free GFP + fusion protein GFP-Atg8), calculated using the Image LabTM version 6.1.0 build 7 Standard edition 2020 Bio-Rad Laboratories, Volume Tool Software after capturing the image with the ChemiDocTM XRS+ System (Bio-Rad, Hercules, CA, USA).

BY4741 *lsm4Δ1-*GFP colonies were tested for *lsm4Δ1-*GFP protein production using Western Blot analysis. Exponentially growing cells (0.2–0.4 OD_600_) in YPD medium were harvested and washed with H_2_O. Then protein extraction, SDS-PAGE and immunoblotting followed the same protocol as already described using the anti-GFP antibody (α-mouse-GFP, Santa Cruz Biotechnology, Dallas, TX, USA, sc-9996) and the secondary antibody HRP-associated anti-mouse (goat) (sc-2060, Santa Cruz Biotechnology, Dallas, TX, USA).

### 4.7. Construction of lsm4Δ1-GFP Genome Mutant Using the CRISPR/Cas9 Editing System

The gRNA of interest was found using the bioinformatic tool Benchling ©, and the *LSM4* gRNA of interest was cloned into the pCfB2311 plasmid using the 5′-phosphorylated primers listed in Table 4 (the gRNA sequence is reported in bold) and the proofreading Velocity DNA Polymerase (Meridian Bioscience, Cincinnati, OH, USA, BIO-21098), swapping the ade2 gRNA already present according to Stovicek et al., 2015 [45]. The obtained plasmid pCfB2311-Lsm4Δ1 was transformed into *E. coli* DH5α cells, made competent using the CaCl_2_ method, and positive cells were selected on LB plates supplemented with 100 mg/L ampicillin. Plasmid DNA was extracted with ISOLATE II Plasmid Mini Kit (Meridian Bioscience, Cincinnati, OH, USA, BIO-52056) and screening of *E. coli* cells was performed through PCR using the forward primer gRNAlsm4contr for (Table 4) complement to Lsm4 gRNA and the reverse primer Sup4rev1 (Table 4) complement to pCfB2311 plasmid. PCR-positive products were purified using the Gel/PCR DNA Fragments Extraction Kit (Geneaid, New Taipei City, Taiwan, DF100) and sequenced in both directions using primers complementary to the plasmid (Sup4rev1 and SNR52for, Table 4). Sequencing was performed by Microsynth AG, Switzerland.

Yeast cells were then transformed with the pCfB2312 plasmid containing the Cas9 gene, according to Gietz and Woods, 2006 [46], plated on YPD supplemented with 200 mg/L G418 sulfate, and incubated at 28 °C for three–four days. The resulting colonies were then double transformed with the pCfB2311-Lsm4Δ1 plasmid and donor DNA obtained by PCR on the pUG35-URA plasmid (primers listed in Table 4) and plated on YPD supplemented with both 200 mg/L G418 sulfate and 100 mg/L nourseothricin. After 4–6 days at 28 °C, the resulting colonies were plated on YPD without antibiotics to allow the loss of plasmids. Screening of colonies was performed through PCR (primers listed in Table 3) and sequencing of PCR products, purified with the Gel/PCR DNA Fragments Extraction Kit (Geneaid, New Taipei City, Taiwan, DF100), and sent to Bio-Fab research s.r.l.

### 4.8. DNA Extraction and PCR Analysis

The production of the Lsm4-GFP donor and evaluation of its correct integration at the genome level were performed using PCR. Colonies to be tested were harvested from YPD plates and resuspended in 100 μL lysis solution (0.2 M LiAc, 1% SDS) and then incubated at 70 °C for 15 min. Then, TE buffer was added, and 300 μL of EtOH 96° was used to precipitate the nucleic acid, and the samples were incubated at −20 °C for 30 min. After centrifugation at 10,000 rpm for 5 min, 3 μL of the supernatant was used as a template for the PCR reaction using Accuzyme ^TM^ DNA polymerase (Meridian Bioscience, Cincinnati, OH, USA, BIO-21052) with primers listed in Table 4. The resulting products were separated by electrophoresis on a 1% agarose gel in TAE buffer 1× (40 mM Tris, 20 mM Acetate, and 1 mM EDTA).

### 4.9. Statistical Analysis

The presented data show the mean of three independent biological experiments, and the error bars represent the standard deviation. For DAPI and DHR 123 analysis, >700 cells were counted per set, and for GFP-Atg8 dots analysis, >300 cells were counted per set.

To evaluate the statistical significance, a two-tailed, two-sample unequal variance test was performed, and the number of stars (*) indicates the *p*-value range: * *p*-value < 0.05, ** *p*-value < 0.01, *** *p*-value < 0.001, **** *p*-value < 0.0001, no star: no statistically significant.

## Figures and Tables

**Figure 1 ijms-27-00464-f001:**
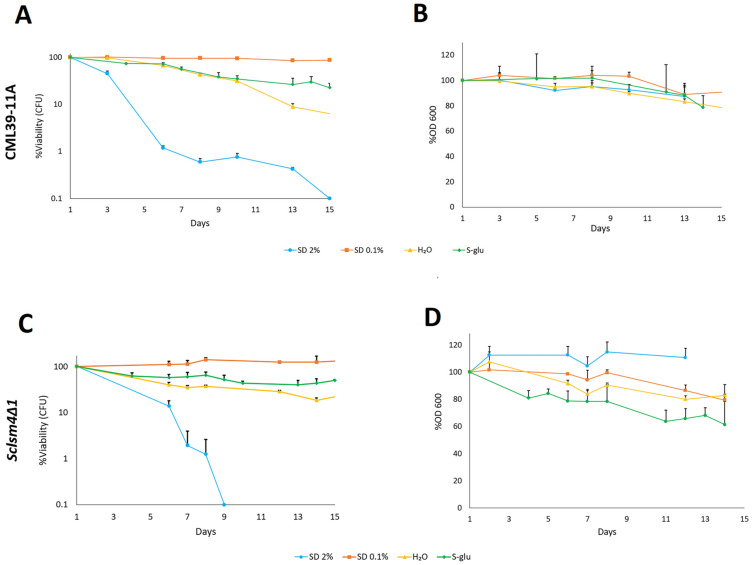
Chronological lifespan of the CML39-11A strain (wild type) (**A**) and *Sclsm4Δ1* strain (mutant) (**C**) under calorie and extreme calorie restriction conditions. The percentage of viability was normalized to day 1. Panels (**B**) (wild type) and (**D**) (mutant) illustrate the percentage of optical density (OD) of cultures relative to day 1. Average and standard deviation of three independent experiments are shown.

**Figure 2 ijms-27-00464-f002:**
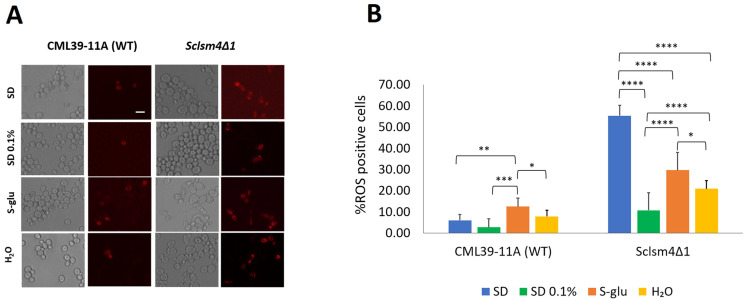
ROS accumulation under calorie restriction in the exponential phase. (**A**) Fluorescence microscopy of DHR 123 stained cells. (**B**) Quantification of % ROS-positive cells. Scale bar: 5 µm. * *p*-value < 0.05, ** *p*-value < 0.01, *** *p*-value < 0.001, **** *p*-value < 0.0001.

**Figure 3 ijms-27-00464-f003:**
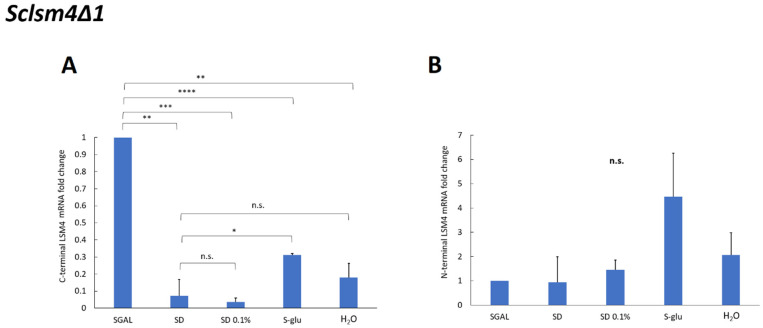
Fold change in C-terminal (**A**) and N-terminal (**B**) of *LSM4* mRNA in different media compared to SGAL-cultured samples. * *p*-value < 0.05, ** *p*-value < 0.01, *** *p*-value < 0.001, **** *p*-value < 0.0001, n.s.: not significant).

**Figure 4 ijms-27-00464-f004:**
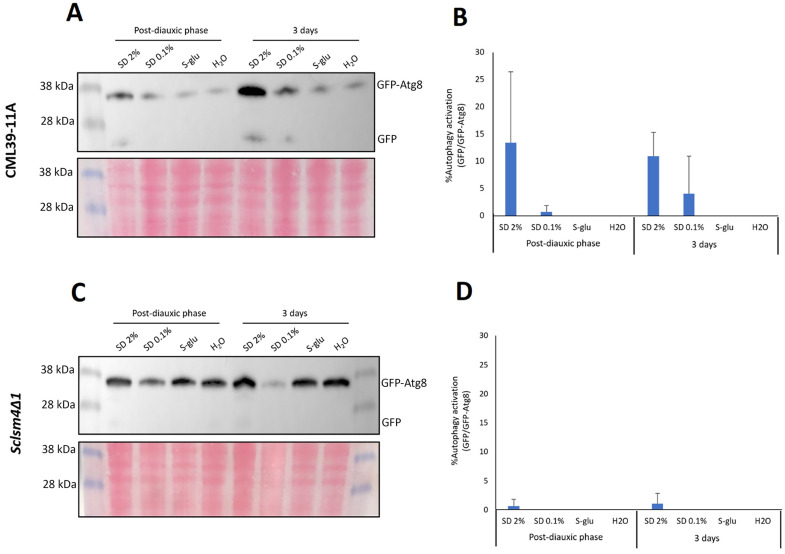
Western blot analysis of autophagy activation under calorie restriction in both wild-type (CML39-11A, panel (**A**)) and mutant (*Sclsm4Δ1*, panel (**C**)) strains. CML39-11A (WT) and mutant *Sclsm4Δ1* cells were cultivated in SD medium during the exponential growth phase. Subsequently, an equal number of cells were washed, resuspended in SD, SD 0.1%, S-glu, and water, and further incubated for either 16 h (Post-diauxic phase) or three days (3 days). Ponceau Red staining was used as a loading control. Autophagy activation was quantified by calculating the ratio of free GFP to GFP-ATG8 for both wild type and *Sclsm4Δ1*mutant (Panels (**B**) and (**D**), respectively).

**Figure 5 ijms-27-00464-f005:**
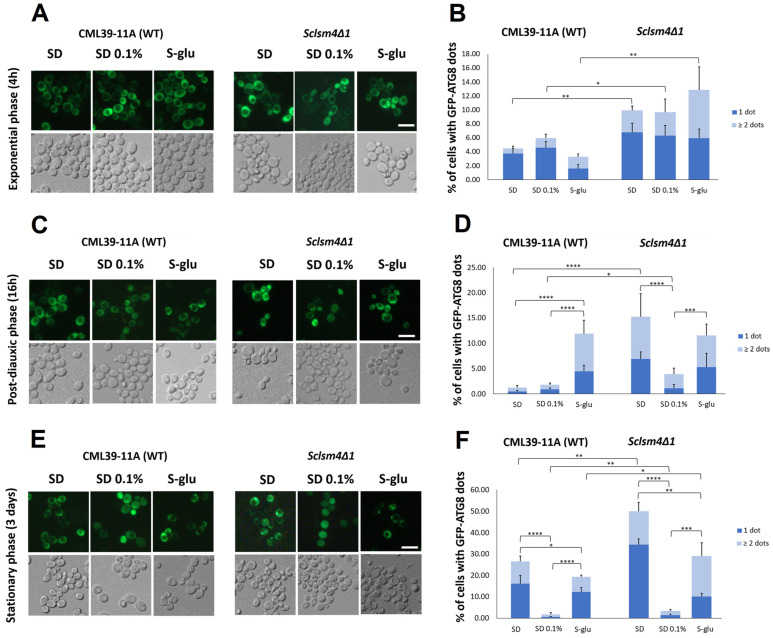
Fluorescent microscopy images were taken of wild-type CML39-11A and MCY4/*Sclsm4Δ1* cells that expressed the GFP-Atg8 fusion protein. These images were captured during the exponential growth phase in SD, SD with 0.1% glucose, and S-glucose media for a duration of 4 h (**A**), during the post-diauxic phase at 16 h (**C**), and after 3 d of growth in either SD or SD with 0.1% glucose and S-glucose (**E**). The number of GFP-Atg8 dots per cell was quantified from three biological replicates (*n* ≥ 300 cells), and the average number of cells with one or more than two dots is displayed in (**B**,**D**,**F**). Scale bar: 5 µm. Error bars represent the standard deviation. * *p*-value < 0.05, ** *p*-value < 0.01, *** *p*-value < 0.001, **** *p*-value < 0.0001.

**Figure 6 ijms-27-00464-f006:**
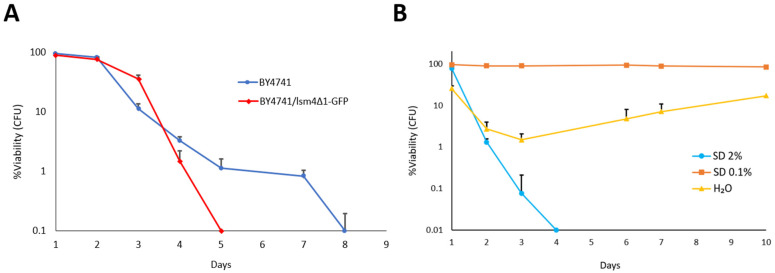
(**A**) Chronological lifespan of BY4741 (wild type strain, in blue) and BY4741/*Sclsm4Δ1*-GFP mutant strain in nutrient-rich conditions (SD 2% glucose, in red). The data are shown as the average of three separate experiments ± standard deviation. (**B**) Chronological lifespan of the *Sclsm4Δ1*-GFP mutant strain under nutrient-rich conditions (SD 2% glucose, light blue), calorie restriction (SD 0.1% glucose, orange), and extreme calorie restriction (water, yellow). Data are presented as the mean of three independent experiments ± standard deviation.

**Table 4 ijms-27-00464-t004:** Primers used for the construction of the *lsm4Δ1-*GFP genome mutant.

Primer Name	Oligonucleotide Sequence	Function
gRNAlsm4For-P	5′-P-GCA AGA TAA TAT AAT TGA CAG TTT TAG AGC TAG AAA TAG CAA GTA AAA TAA GGC-3′	Cloning of *LSM4* gRNA in pCfB2311 plasmid—Forward
SNR52Rev-P	5′-P-GAT CAT TTA TCT TTC ACT GCG GCG AAG-3′	Cloning of *LSM4* gRNA in pCfB2311 plasmid—Reverse
gRNAlsm4contr for	5′-GAA AGA TAA ATG ATC GCA AGA TAA TAT AAT TGA CA-3′	Screening of transformed *E. coli*—Forward
Sup4rev1	5′-CCC CCG CTA GCG CGT TGT AAA ACG ACG GCC AGT G-3′	Screening of transformed *E. coli*—Reverse Primer used for plasmid sequencing
SNR52for	5′-CCC CCG AAT TCG AGC GGA TAA CAA TTT CAC ACA GG-3′	Primer used for plasmid sequencing
Lsm4-GFP For	5′-GGG ACT TTT ATC ATC AAG TTT ATC AAA TTG CAA GAT AAT ATA ATC GAT ACC GTC GAC CTC GAC-3′	Production of donor Lsm4-GFP cassette—Forward
Lsm4-GFP Rev	5′-GGG CCG TTA CTA TTA GAG TTA TTG TTG GAG TTA ATT TGC TGA CGT TGT AAA ACG ACG GCC-3′	Production of donor Lsm4-GFP cassette—Reverse
ScLSM4 for	5′-AAA AAA GGA TCC GTA CGC AGT CAC AAT GCG G-3′	Screening of transformed yeast colonies—Forward
ScLSM4 rev	5′-TTG GAC GGA CCC ACC TAA AC-3′	Screening of transformed yeast colonies—Reverse

## Data Availability

The original contributions presented in this study are included in the article/Appendix A. Further inquiries can be directed to the corresponding author.

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
