# Peer review of "Calorie Restriction Suppresses Premature Ageing in Pro-Apoptotic Yeast Mutants Through an Autophagy-Independent Mechanism"

_ijms, 2026, doi:10.3390/ijms27010464_

Round 1
Reviewer 1 Report
Comments and Suggestions for Authors
Comment to the Author
In this article, Benedetta Caraba et al. use the yeast Saccharomyces cerevisiae to investigate how the calorie restriction can suppress ageing in pro-apoptotic yeast mutant. They apply different calorie restriction conditions, ranging from the standard repressive glucose concentration (2%) to an extreme condition with only water, evaluating their effects on a yeast strain in which the wild type chromosomal copy of the gene is under control of GAL promoter and the truncated mutant copy is expressed from its endogenous promoter on a centromeric plasmid. Under calorie restricted conditions, they report an extended chronological lifespan accompanied by reduced ROS levels, which are not due to altered gene expression or increased autophagic flux. These observations appear independent of the yeast genetic background, as the same phenotype was also observed in a BY4741 mutant strain generated via CRISPR-Cas9. The study is undoubtedly interesting; however, several concerns must be addressed to improve its scientific robustness.
MINOR REVISION
1) Line 18 as 44: use italic formatting for Saccharomyces cerevisiae.
2) Line 40: after “reactive oxygen species”, add (ROS) and remove the specification at line 149.
3) Line 44, 224, 235..etc: replace all incorrected forms of Sclsm4D1 with the correct notation Sclsm4Δ1
MAJOR REVISION
-The authors use four nutrient conditions: Synthetic medium plus 2% glucose, Synthetic medium plus 0,1% glucose, Synthetic medium without glucose and water. Synthetic medium without glucose is already an extreme condition, since cells lack any carbon source. Moreover, the results obtained under the water condition suggest that the strain survives for 15 days without carbon, nitrogen, phosphate, or protein sources.
In panels B and D, is it possible to include OD values at day 1? How many cell divisions occur for the wild-type and mutant strains when diluted the day before to a final concentration of 5×10⁵ cells/mL?
- The authors use the MCY4/lsm4Δ1 strain, in which the chromosomal LSM4 copy is under the GAL1-GAL10 promoter, and the truncated version is on a centromeric plasmid. At line 225, they state that the chromosomal copy is not activated under calorie-restriction conditions. This should be highlighted more clearly when demonstrated.
Furthermore, how can the significant increase in C-terminal LSM4 mRNA fold-change under the S-glu condition (Figure 3, Panel A) in the Sclsm4Δ1 strain be explained?
Why did the authors not choose a clearer genetic background, such as a complete knockout strain expressing only the truncated version from a centromeric plasmid?
- Fig4. The authors use Ponceau Red staining as a loading control to assess autophagy activation. Although it is useful to evaluate transfer efficiency or overall protein loading on the membrane, it is generally not considered a reliable loading control. There are many available antibodies for yeast housekeeping proteins such as POR or PGK; it is recommended to confirm these results using such controls
-Figures S2 and S3 report data obtained with the BY4741 mutant strain. In lines 230–236, the authors compare this strain with the previously characterized mutant strain. Would it be possible to compare these strains directly in the same experiments and prepare new figures presenting their phenotypes side-by-side under the same conditions?
Author Response
We would like to thank the reviewer for the useful suggestions and observations. Here below is a point-by-point response:
1) Line 18 as 44: use italic formatting for Saccharomyces cerevisiae.
Done
2) Line 40: after “reactive oxygen species”, add (ROS) and remove the specification at line 149.
Done
3) Line 44, 224, 235..etc: replace all incorrected forms of Sclsm4D1 with the correct notation Sclsm4Δ1
Done
4) The authors use four nutrient conditions: Synthetic medium plus 2% glucose, Synthetic medium plus 0,1% glucose, Synthetic medium without glucose and water. Synthetic medium without glucose is already an extreme condition, since cells lack any carbon source. Moreover, the results obtained under the water condition suggest that the strain survives for 15 days without carbon, nitrogen, phosphate, or protein sources.
In panels B and D, is it possible to include OD values at day 1? How many cell divisions occur for the wild-type and mutant strains when diluted the day before to a final concentration of 5×10⁵ cells/mL?
We thank the reviewer for this observation. We have now clarified the experimental procedure in the Materials and Methods section. Moreover, we have added a table to the supplemental material (Table SM1) reporting the OD₆₀₀ values at day 1 for each strain and condition.
5) The authors use the MCY4/lsm4Δ1 strain, in which the chromosomal LSM4 copy is under the GAL1-GAL10 promoter, and the truncated version is on a centromeric plasmid. At line 225, they state that the chromosomal copy is not activated under calorie-restriction conditions. This should be highlighted more clearly when demonstrated.
The fact that the chromosomal copy of LSM4 is not reactivated under calorie-restriction conditions is demonstrated by the qPCR transcript analysis shown in Figure 3. We thank the reviewer for this observation; to clarify this point more explicitly, we have added a sentence to the Discussion (lines 2278-281).
6) Furthermore, how can the significant increase in C-terminal LSM4 mRNA fold-change under the S-glu condition (Figure 3, Panel A) in the Sclsm4Δ1 strain be explained?
The absence of glucose leads to derepression of the GAL1–GAL10 promoter; however, full induction still requires the presence of galactose. The low level of transcript detected under glucose-free conditions may therefore reflect the promoter’s basal activity. In any case, since the reported phenotypes are also observed in SD medium containing 0.1% glucose, it is reasonable to hypothesize that the extension of chronological lifespan under SD-glu and H₂O conditions cannot be attributed to reactivation of the GAL1–GAL10 promoter and to the presence of the full-length LSM4 transcript.
7) Why did the authors not choose a clearer genetic background, such as a complete knockout strain expressing only the truncated version from a centromeric plasmid?
LSM4 is an essential gene. The MCY4 strain was extensively studied in previous work and is therefore very well characterized.
Moreover, because expression of the truncated protein is highly mutagenic (see Stirpe et al., 2017), maintaining stable expression of the truncated variant in a complete knockout background over extended periods is challenging, at least in this genetic background.
Nevertheless, we have now generated a new strain that carries the truncated form of the protein in a different genetic background. We are currently investigating whether the absence of the C-terminal region makes the strain more susceptible to acquiring mutations. Apparently, phenotypes in this genetic background (BY4741) are still present but less pronounced compared to the MCY4 background.
8) Fig4. The authors use Ponceau Red staining as a loading control to assess autophagy activation. Although it is useful to evaluate transfer efficiency or overall protein loading on the membrane, it is generally not considered a reliable loading control. There are many available antibodies for yeast housekeeping proteins such as POR or PGK; it is recommended to confirm these results using such controls
We agree with the reviewer that Ponceau Red alone cannot serve as a loading control, but only as a control for protein transfer onto the membrane. In the experiments shown in Figure 4, the focus is not on comparing absolute expression levels of the GFP–Atg8 fusion protein, but rather on assessing the ratio between free GFP—released during autophagy—and the intact GFP–Atg8 protein. Because this analysis relies on a within-sample comparison, normalization to a housekeeping protein such as PGK or TUB1 would not provide additional meaningful information for the interpretation of the results.
9) Figures S2 and S3 report data obtained with the BY4741 mutant strain. In lines 230–236, the authors compare this strain with the previously characterized mutant strain. Would it be possible to compare these strains directly in the same experiments and prepare new figures presenting their phenotypes side-by-side under the same conditions?
The phenotypes of the MCY4/Sc lsm4Δ1 mutant were previously described in Figure 1 of Caraba et al., 2023. The experimental conditions used here are identical to those applied in all our studies on programmed cell death phenotypes in strains carrying LSM4 mutations. Each experiment included appropriate controls, represented in this case by the wild-type BY4741 strain.
Moreover, to ensure that GFP tagging did not introduce unwanted effects, we included an additional control strain expressing the full-length LSM4–GFP fusion. As shown in Figure 2 of the Supplementary Material, the BY4741/lsm4Δ1–GFP strain exhibits increased nuclear fragmentation and ROS production, particularly in stationary phase. When comparing these absolute values with those reported in Caraba et al., 2023, the phenotype—although clearly detectable—appears to be less pronounced.
These differences may stem from the distinct genetic backgrounds used, as it is well established that the impact of a mutation can vary substantially depending on the genomic context. We are currently assessing whether this new strain, BY4741/lsm4Δ1–GFP, also displays genetic instability, as described for the MCY4 background. This analysis will help clarify the extent to which genetic background influences the manifestation of apoptotic and aging phenotypes.
Reviewer 2 Report
Comments and Suggestions for Authors
1.line34-38: need a reference.
2.Result:The lack of significant GFP-Atg8 processing via Western blot (Fig. 4) and a reduction in GFP-Atg8 puncta via microscopy under moderate CR (Fig. 5). However, these data only show a lack of autophagosome accumulation. They do not rule out the possibility of an increased or highly efficient autophagic flux, where autophagosomes are formed and rapidly degraded in the vacuole.
3.In Figure 1D, the authors note an ~20% cell lysis for the Sclsm4Δ1 mutant in S-glu medium. This is a significant loss of culture integrity. It is not clear how this lysis might confound the CLS data presented in Figure 1C, which is measured by colony-forming ability. The discussion should address whether this lysis could artificially influence the perceived viability or if the CLS is underestimated in this condition.
4.Throughout the manuscript, species and gene names are not consistently italicized. Please ensure standard nomenclature is followed.
5.There appear to be typos or inconsistencies in the naming of the mutant. Please ensure consistency.
Comments on the Quality of English LanguageThere are some sentences in the introduction and discussion sections of the manuscript that need to be integrated and the logic needs to be reorganized.
Author Response
We thank the reviewer for the helpful comment and suggestions. Here below is a point-by-point response:
1.line34-38: need a reference.
We thank the reviewer for the suggestion, we added some references
2.Result: The lack of significant GFP-Atg8 processing via Western blot (Fig. 4) and a reduction in GFP-Atg8 puncta via microscopy under moderate CR (Fig. 5). However, these data only show a lack of autophagosome accumulation. They do not rule out the possibility of an increased or highly efficient autophagic flux, where autophagosomes are formed and rapidly degraded in the vacuole.
We thank the reviewer for the observation. We added a sentence in the discussion that take into consideration also this possibility (lines 293-296).
3.In Figure 1D, the authors note an ~20% cell lysis for the Sclsm4Δ1 mutant in S-glu medium. This is a significant loss of culture integrity. It is not clear how this lysis might confound the CLS data presented in Figure 1C, which is measured by colony-forming ability. The discussion should address whether this lysis could artificially influence the perceived viability or if the CLS is underestimated in this condition.
We thank the reviewer for the observation. We added a sentence in the discussion on the possible role of cell lysis ( lines 268-271).
4.Throughout the manuscript, species and gene names are not consistently italicized. Please ensure standard nomenclature is followed.
Done
5.There appear to be typos or inconsistencies in the naming of the mutant. Please ensure consistency.
Done
Comments on the Quality of English Language
There are some sentences in the introduction and discussion sections of the manuscript that need to be integrated and the logic needs to be reorganized.
We reorganized the text as suggested
Round 2
Reviewer 1 Report
Comments and Suggestions for Authors
The authors’ responses are clear and comprehensive, and they adequately address all the points raised. I have no further comments on this matter.
Author Response
We thank the reviewer for the helpful comments and for having accepted our changes.
Reviewer 2 Report
Comments and Suggestions for Authors
- The conclusion would be significantly strengthened by providing a mechanism for the ROS reduction, rather than just observing it.
- Line 67: "This indicates that Lsm complex play a vital role for the in regulating autophagy." The phrase "for the in" is incorrect.
- Line 80: :"mRNAs,," (double comma).
- Line 90-93: The description of the TORC1 signaling cascade is standard, but please ensure the references cited here are the most current reviews, as this pathway is rapidly evolving.
- Line 174-175: The text states there is a "slight activation" of the full-length gene in S-Glu. Although the authors dismiss this, the error bar in Figure 3A (S-Glu) is large. Is this "slight activation" biologically relevant?
- Line 297: "...induced. . These observations..."
- The reference list contains a significant number of older citations (pre-2010). The authors should ensure they are citing the most recent literature regarding TORC1, autophagy, and yeast aging to demonstrate the current relevance of the study.
Comments on the Quality of English Language
There are some sentences in the introduction and discussion sections of the manuscript that need to be integrated and the logic needs to be reorganized.
Round 2
1. The conclusion would be significantly strengthened by providing a mechanism for the ROS reduction, rather than just observing it.
Thanks for this observation. A possible mechanism underlying the reduction of ROS observed during calorie restriction may involve enhanced mitochondrial efficiency and we discuss this in lines 326-338. Calorie restriction is known to promote mitochondrial remodelling, improved respiratory function, and reduced electron leakage, contributing to lower ROS production in the presence of functional mitochondria. In line with these observations, we previously demonstrated that L-acetyl-Lcarnitine (L-ALC) extends yeast lifespan by improving mitochondrial function and dynamics, but this was not true in their isogenic rho0 mutant strains (V. Palermo, C. Falcone, M. Calvani, C. Mazzoni, Acetyl-L-carnitine protects yeast cells from apoptosis and aging and inhibits mitochondrial fission, Aging Cell 9 (2010) 570–579. https://doi.org/10.1111/j.1474-9726.2010.00587.x.). These findings suggest that similar mitochondrial-centered mechanisms may operate under calorie restriction, ultimately leading to reduced oxidative stress and improved chronological lifespan.
2. Line 67: "This indicates that Lsm complex play a vital role for the in regulating autophagy." The phrase "for the in" is incorrect.
Thanks for the observation it was a typo and it has been corrected
3. Line 80: :"mRNAs,," (double comma).
Thanks for the observation it was a typo and it has been corrected
4. Line 90-93: The description of the TORC1 signaling cascade is standard, but please ensure the references cited here are the most current reviews, as this pathway is rapidly evolving.
We thank the reviewer for this observation. We updated the reference with more recent once
5. Line 174-175: The text states there is a "slight activation" of the full-length gene in S-Glu. Although the authors dismiss this, the error bar in Figure 3A (S-Glu) is large. Is this "slight activation" biologically relevant?
We thank the reviewer for this observation. When compared to SD, the t-test yields a p-value of 0.049, which is conventionally considered statistically significant (p < 0.05). However, despite this statistical significance, the magnitude of the difference is small and likely of limited biological relevance, particularly in light of the variability observed in S-Glu conditions. For the sake of clarity and transparency, we decided to discuss this result in the manuscript.
6. Line 297: "...induced. . These observations..."
Thanks for the observation it was a typo and it has been corrected
7. The reference list contains a significant number of older citations (pre-2010). The authors should ensure they are citing the most recent literature regarding TORC1, autophagy, and yeast aging to demonstrate the current relevance of the study.
We thank the reviewer for this observation. We updated the references with more recent ones, although some older references remain important for the field and merit being cited.
Round 3
Reviewer 2 Report
Comments and Suggestions for Authors
The manuscript contains "orphan" paragraphs consisting of a single sentence or very few lines, which disrupts the reading flow and appears unprofessional.
The logic currently jumps back and forth between the specific features of the lsm4 mutant and broad, textbook-level descriptions of nutrient signaling pathways, causing the reader to lose the central thread of the study.
Comments on the Quality of English LanguageThere are some sentences in the introduction and discussion sections of the manuscript that need to be integrated and the logic needs to be reorganized.
Author Response
- The manuscript contains "orphan" paragraphs consisting of a single sentence or very few lines, which disrupts the reading flow and appears unprofessional.
Thanks for this suggestion, we tried to avoid this. As an example, here below some change we made
Lines 91-94: fused two sentences in one
Lines 101-102: this paragraph was fused to the previous one
Lines 107-110: two single sentences were fused
Lines 128-146: we rephrased the description of TORC1 regulation in nitrogen and glucose starvation.
Lines 149-153: two single sentences were fused
Lines 158-160: two single sentences were fused
Lines 218-220: two single sentences were fused
Lines 223-227: two single sentences were fused
- The logic currently jumps back and forth between the specific features of the lsm4 mutant and broad, textbook-level descriptions of nutrient signaling pathways, causing the reader to lose the central thread of the study.
We clustered Lsm4 information, and the nutrient signaling pathway description was simplified, indicating the references to a more specialized description of these complicated pathways.
- There are some sentences in the introduction and discussion sections of the manuscript that need to be integrated and the logic needs to be reorganized.
We reorganized both the introduction and discussion. In this section, we concentrated on the possible role of the Lsm complex in the intricate pathway of nutrient signaling and autophagy.
The discussion was divided in three paragraphs:
3.1. Calorie restriction mitigates oxidative stress and extends lifespan in the Sclsm4Δ1 mutant
Here we discussed that Calorie restriction (CR) significantly extends Sclsm4Δ1 mutant lifespan and reduces ROS accumulation without altering the genomic GAL1-10-LSM4 gene expression. CR improved cellular fitness by modulating metabolic and oxidative balance
3.2. Calorie restriction does not activate autophagy in either wild-type or mutant strains
Here we discussed about the lack of autophagic flux during calorie restriction (CR) suggesting that CR’s protective effects on lifespan and ROS reduction occur independently of canonical autophagy
3.3. Calorie restriction restores lifespan in the genomic Sclsm4Δ1 mutant
Calorie restriction (CR) is active in protecting from ageing linked to Sclsm4Δ1 mutant, also in a genomic copy constructed by CRISPR-CAS9 in a different genetic background
In the “Concluding remarks” although, for the moment, the molecular mechanisms remain elusive, we also propose some hints on the possible mechanisms, pointing to metabolism and mitochondrial efficiency.